# Anti-necroptotic effects of human Wharton's jelly-derived mesenchymal stem cells in skeletal muscle cell death model via secretion of GRO-α

**Sang Eon Park**[1,2☯], **Soo Jin Kwon**[1,2☯], **Sun Jeong Kim**[1,2☯], **Jang Bin Jeong**[1,2], **Min-Jeong Kim**[2], **Suk-joo Choi**[3], **Soo-young Oh**[3], **Gyu Ha Ryu**[4,5], **Hong Bae Jeon**[1]*, **Jong Wook Chang**[1,2,6]*

1 Cell and Gene Therapy Institute, ENCell Co. Ltd, Seoul, Republic of Korea, 2 Cell and Gene Therapy Institute, Samsung Medical Center, Seoul, Republic of Korea, 3 Department of Obstetrics and Gynecology, Samsung Medical Center, Seoul, Republic of Korea, 4 Department of Medical Device Management and Research, Samsung Advanced Institute for Health Sciences & Technology, Sungkyunkwan University School of Medicine, Seoul, Republic of Korea, 5 The Office of R&D Strategy & Planning, Samsung Medical Center, Seoul, Republic of Korea, 6 Department of Health Sciences and Technology, Samsung Advanced Institute for Health Sciences & Technology, Sungkyunkwan University, Seoul, Korea

☯ These authors contributed equally to this work.

\* jwchang@encellinc.com, jongwook.chang@samsung.com (JWC); jhb@encellinc.com (HBJ)

**Data Availability Statement:** All data that support the findings of this study are within the manuscript and its Supporting information files. The original uncropped and unadjusted images underlying all

## Abstract

Human mesenchymal stem cells (hMSCs) have therapeutic applications and potential for use in regenerative medicine. However, the use of hMSCs in research and clinical medicine is limited by a lack of information pertaining to their donor-specific functional attributes. In this study, we compared the characteristics of same-donor derived placenta (PL) and Wharton's jelly (WJ)-derived hMSCs, we also compared their mechanism of action in a skeletal muscle disease *in vitro* model. The same-donor-derived hWJ- and hPL-MSCs exhibited typical hMSC characteristics. However, GRO-α was differentially expressed in hWJ- and hPL-MSCs. hWJ-MSCs, which secreted a high amount of GRO-α, displayed a higher ability to inhibit necroptosis in skeletal muscle cells than hPL-MSCs. This demonstrates the anti-necroptotic therapeutic effect of GRO-α in the skeletal muscle cell death model. Furthermore, GRO-α also exhibited the anti-necroptotic effect in a Duchenne muscular dystrophy (DMD) mouse model. Considering their potential to inhibit necroptosis in skeletal muscle cells, hWJ-MSCs and the derived GRO-α are novel treatment options for skeletal muscle diseases such as DMD.

## Introduction

Human mesenchymal stem cells (hMSCs) are multipotent somatic stem cells that can be derived from a wide range of tissues, including the adipose tissue (AT), bone marrow (BM), umbilical cord blood (UCB), Wharton's jelly (WJ), placenta (PL), and dental pulp [1–5].

blot or gel results were reported in S4 Fig. The antibody array data generated in the present study can be found in the GEO under accession number GSE277126 (https://www.ncbi.nlm.nih.gov/geo/query/acc.cgi?acc=GSE277126).

**Funding:** This study was supported by grants awarded by the Korea Health Technology R&D Project through the Korea Health Industry Development Institute (KHIDI) (HR22C1363 and HI14C3484), and funded by the Ministry of Health & Welfare, Republic of Korea. This work was supported by the Collabo R&D between Industry, Academy, and Research Institute (S3098634) funded by the Ministry of SMEs and Startups (MSS, Korea), by grants from the National Research Foundation of Korea (2021R1F1A1064060 and 2021R1F1A1062366), and supported by Korean Fund for Regenerative Medicine funded by Ministry of Science and ICT, and Ministry of Health and Welfare (RS-2022-00060268, RS-2023-0023069, Republic of Korea), and by Samsung Medical Center (SMO1220041). The funding bodies played no role in the design of the study and collection, analysis, and interpretation of data and in writing the manuscript.

**Competing interests:** The authors have declared that no competing interests exist.

hMSCs have the potential to differentiate into cells of the mesodermal lineages and produce trophic factors that enable migration and proliferation in response to cytokines or chemokines released from the site of injury for tissue repair and regeneration [6–8]. Their ease of isolation, culture expansion, multi-potential differentiation, and immunomodulatory properties underly the potential of hMSCs in regenerative medicine, in addition to their existing therapeutic applications [9–12]. Therefore, MSCs have been evaluated in several clinical trials for currently untreatable diseases, such as skeletal muscle diseases [13], bone and cartilage defects [14], myocardial infarction [15], and autoimmune diseases [16]; these trials have demonstrated the potential of hMSCs with no serious adverse events [4,17,18].

Despite the therapeutic effects, the use of hMSCs in research and clinical medicine is limited by the lack of information pertaining to their functional attributes that differ with tissues [19–21]. Several comparative analyses on mesenchymal stem cells derived from different sources have been conducted [22–25]. Many studies have reported that birth-associated hMSCs have biological advantages over adult sources [26]. Additionally, our group previously reported that the therapeutic secretome of birth-associated tissue-derived hMSCs, such as WJ- and PL-derived hMSCs, is more diverse than that of AT-and BM-derived hMSCs [27]. However, this analysis was limited because the samples were not obtained from the same donor.

In this study, we focused on hMSCs derived from birth-associated tissues, hWJ- and hPL-MSCs, from the same donors. We compared and analyzed the biological characteristics of hMSCs concerning cell morphology, growth rate, immunophenotype, and differentiation potential under the same conditions. To elucidate the anti-cell death mechanism of tissue-specific hMSCs derived from the same donor, we compared the same donor-derived hWJ- and hPL-MSCs; in addition, we analyzed the skeletal muscle cell death inhibitory mechanisms of these cells in the *in vitro* and *in vivo* skeletal muscle cell death models.

We identified growth-regulated protein alpha (GRO-α) as the key hWJ-MSC-derived paracrine factor mediating the anti-necroptosis signaling pathway in the skeletal muscle cell death model.

## Materials and methods

### Sample collection

In accordance with the guidelines approved by the Institutional Review Board (IRB) of Samsung Medical Center, umbilical cords and PL were collected with informed consent from pregnant mothers (IRB#2016-07-102). Samples were collected from April 3rd, 2017, and continued until August 13th, 2019, to investigate the optimal selection of mesenchymal stem cells by source for the treatment of chronic-incurable diseases.

### Co-culture of C2C12 cells with hMSCs

hMSCs were isolated from WJ and PL, as previously reported [27,28]. All hMSCs were used at passage six. C2C12, a mouse myoblast cell line (ATCC® CRL-1772™, Rockville, MD, USA), was cultured as previously reported in [28]. When C2C12 reached 80–90% confluency after seeding on a six-well plate (BD Falcon, Franklin Lakes, NJ, USA), they were treated with 10 μM lovastatin for 3 h to induce cell death and washed with PBS to remove lovastatin. hWJ- and hPL-MSCs were trypsinized, suspended in serum-free MEM alpha, seeded ($1\times10^5$/1 mL) in 1-μm-pore-size transwell inserts (BD Biosciences, Franklin Lakes, NJ, USA), and co-cultured with C2C12 for 24 h. C2C12 myoblasts ($2\times10^5$ cells/ 2 mL) were treated with or without human recombinant GRO-α (0–200 ng/mL) (R&D Systems, Minneapolis, MN, USA) for the indicated time periods in serum-free DMEM.

## Human WJ-MSCs and GRO-α injection into mdx mice

Mdx mice are spontaneous mutants that do not express dystrophin, and therefore are a suitable model of Duchenne muscular dystrophy (DMD) [29]. Mdx (C57BL/10ScSn-Dmdmdx/J; JAX#001801) and wild-type (C57BL/10ScSnJ; JAX#000476) mice were purchased from Jackson Laboratory (Bar Harbor, ME, USA). Offspring mice were bred and maintained according to the protocols recommended by Jackson Laboratory.

A total of 15 Two- to five- month-old mdx male mice and 5 five-month-old wild-type mice were placed in a restraint device and a heating pad (37˚C) was placed over the tails to dilate the tail veins. Mdx mice were injected with PBS (control, n = 5), hWJ-MSCs ($5\times10^4$ cells/ 100 μL suspended in PBS, n = 5), or GRO-α (1–100 μg/ 100 μL suspended in PBS, n = 5) through the lateral tail vein using an insulin syringe. For minimize potential confounders, hWJ-MSCs, GRO-α and PBS administration were alternated. Also, no criteria were set and no experimental units or data points were used for data analysis.

The mice were euthanized using a carbon dioxide-only chamber in the animal experiment center of Samsung Medical Center. Euthanasia was performed on the $7^{th}$ day after injection. The mice were placed in a transparent chamber, the carbon dioxide cylinder valve was opened, and $CO_2$ was injected at a low flow rate (20–30% of cage volume per minute); this was continued until the animals stopped breathing.

## Creatine kinase activity (CK assays)

Serum was collected from blood and creatine kinase (CK) activity assay was performed as previously described [29]. CK activity was determined using the Creatine Kinase Activity Assay Kit (Colorimetric) (ab155901, Abcam, Cambridge, UK).

## Cell counting kit-8 (CCK-8) assay

The viability of C2C12 cells was measured using the Cell Counting kit-8 (CCK-8, Dojindo, Tokyo, Japan), as previously described [30]. The absorbance at 450 nm was determined using a microplate reader (xMark™ Microplate Absorbance Spectrophotometer Bio-Rad Laboratories, Inc.; Hercules, CA, USA).

## Flow cytometric analysis

To confirm the stemness of hWJ- and hPL-MSCs, cell surface marker analysis was performed. Harvested hMSCs were washed in PBS supplemented with 2% FBS to block non-specific binding sites. Immunophenotypic analysis of hMSCs was performed using flow cytometry for the following markers: CD44, CD73, CD90, CD105, CD11b, CD14, CD19, CD34, CD45, and HLA-DR (MHCII) (BD Biosciences, USA). At least 10,000 events were acquired using the BD FACS Verse flow cytometer (BD Biosciences). The results were analyzed using the BD FAC-Suite software version 10. Flow cytometry was also performed for appropriate isotype controls.

## Trilineage differentiation

Another method to test stemness is to examine the differentiation capabilities of hMSCs. Trilineage differentiation assays were performed, as previously reported [27].

## Western blotting

Western blotting analysis was performed as previously described [31]. The membranes were blocked with 5% skim milk in TBS-T (containing 0.2% Tween-20) at room temperature with gentle shaking for 60 min, followed by incubation with primary antibodies (1:1,000–1:5,000

dilutions in 5% skim milk) in a cold room (4˚C) overnight, with gentle shaking. After washing thrice with TBS-T (10 min/wash), the membranes were incubated with 1:5,000–1:10,000-diluted HRP-conjugated anti-rabbit (abc-5003) or anti-mouse (abc-5001) secondary antibodies in 5% skim milk at room temperature for 1 h with gentle shaking. The membranes were washed thrice (10 min/wash) and treated with ECL solution (Bio-Rad Laboratories) for 30 s before detecting the bands. The original images underlying all western blotting results can be confirmed in S1 Fig.

## Antibody arrays

Analysis using antibody arrays was performed as previously reported [29]. The RayBio Biotin Label-based Human Antibody Array (#AAH-BLG-1-4), which can detect a total of 507 human proteins, was used for analyzing the secreted proteins in the conditioned media collected from the co-cultivation of hMSCs and C2C12 cells.

## siRNA transfection

The siRNAs against CXCL1 were designed and synthesized at Bioneer (Republic of Korea). WJ-MSCs were transfected with siRNAs using Lipofectamine RNAiMax (Invitrogen) according to the manufacturer's instructions. The WJ-MSC cultures were transfected with negative control and CXCL1 siRNAs (Bioneer, Daejeon, South Korea) for 24 h. The knockdown efficiency was determined through qRT-PCR and ELISA.

## RNA preparation and qRT-PCR

Total RNA was prepared from the cells using TRIzol reagent (Ambion-Thermo Fisher Scientific, Carlsbad, CA). Complementary DNA (cDNA) was synthesized from total RNA using SuperScript (Invitrogen, Carlsbad, CA). qRT-PCR was performed with cDNA in a reaction mix containing the respective primers and the SYBR green master mix (Applied Biosystems-Thermo Fisher Scientific). The PCR conditions were as follows: 40-cycle amplification at 94˚C for 30 s, 60˚C for 40 s, and 72˚C for 1min. The specific primer sets used were as follows: CXCL1, 5′-CCCAAGAACATCCAAAGTGTG-3′ and 5′-CATTCTTGAGTGTGGCTATGAC-3′. Results were normalized against human GAPDH levels and quantified by fitting the threshold cycle (ΔΔCT) values to the standard curve.

## Enzyme-linked immunosorbent assay

The Human GRO-α Quantikine ELISA kit (#DGR00B; R&D systems, USA) was used to analyze the expression of proteins selected from the antibody array. Protein concentration was calculated from the intensity value measured at 450 nm.

## Apoptosis/Necrosis assay

C2C12 cells ($1\times10^5$) were seeded in 6-well plates for 24 h. Subsequently, C2C12 cells treated with lovastatin were either co-cultured with hWJ- and hPL-MSCs or treated with GRO-α for 24 h. Apoptosis and necrosis were determined using the Apoptosis/Necrosis Detection kit (#ab176749; Abcam), according to the manufacturer's instructions.

## Antibodies and reagents

The following primary antibodies were used: poly ADP ribose polymerase (9542L, PARP; polyclonal, Cell Signaling Technology, Danvers, MA, USA), phosphoreceptor interacting protein kinase 3 (57220S, P-RIP3; polyclonal, Cell Signaling Technology), phospho-mixed lineage

kinase domain-like protein (37333S, P-MLKL; monoclonal (D6E3G), Cell Signaling Technology), cleaved Caspase-3 (9661S, polyclonal, Cell Signaling Technology), and beta-actin (sc-47778, monoclonal (C4), Santa Cruz Biotechnology, Santa Cruz, CA, USA). Lovastatin (>98% purity, mevinolin; Sigma-Aldrich) was prepared as a 25 mM stock solution in dimethyl sulfoxide (DMSO) (Sigma-Aldrich), and RIP kinase inhibitor (GSK'872; R&D Systems, USA) was prepared as a stock solution. These reagents were further diluted in DMSO prior to cell treatment. A 100 μg/mL stock solution of Human GRO-α recombinant protein was prepared in PBS, containing at least 0.1% human or bovine serum albumin.

## Ethics approval and consent to participate

This study was approved by the Institutional Animal Care and Use Committee (IACUC) of the Samsung Biomedical Research Institute (SBRI), Samsung Medical Center (SMC) (#20200811001). SBRI is an accredited facility of the Association for the Assessment and Accreditation of Laboratory Animal Care International (AAALAC International). It abides by the Institute of Laboratory Animal Resources (ILAR) guidelines. In accordance with the guidelines approved by the Institutional Review Board (IRB) of Samsung Medical Center, umbilical cords and placenta were collected with informed consent from pregnant mothers (IRB#2016-07-102).

## Statistical analyses

Data are presented as mean ± standard error (SE). Statistical significance was determined using $t$-test and one-way ANOVA, followed by Tukey's post hoc test (*$P < 0.05$, **$P < 0.01$, ***$P < 0.001$). GraphPad Prism (ver. 10) for Windows was used for all analyses.

## Results

### No difference in characteristics of human WJ- and PL-MSCs derived from the same donor

We compared the morphology and surface markers of hWJ- and hPL-MSCs from the same donor. In passage 5, both hWJ- and hPL-MSCs exhibited spindle-shaped morphology (Fig 1A). FACS Verse flow cytometer (FACS) analysis revealed that all hMSCs were positive for the following markers: cluster of differentiation, CD44, CD73, CD90, and CD105, but negative for CD11b, CD14, CD19, CD34, CD45, or HLA-DR, according to the ISCT criteria (S1 Table). To investigate the differentiation capacity of various hMSCs, cells were cultured in osteogenic, adipogenic, and chondrogenic differentiation media. The three-lineage potential for osteogenesis, chondrogenesis, and adipogenesis is a standard for defining multipotent hMSCs. The tri-lineage potential in the hMSCs was tested by staining for typical lineage markers. Osteogenesis was defined by a bone-type marker, determined through alkaline phosphatase (ALP) staining (Fig 1B); adipogenesis was observed through staining of cytoplasmic lipid droplets with Oil Red O (Fig 1C); chondrogenesis, characterized by an increase in proteoglycans, was evaluated using safranin O staining (Fig 1D). These results confirm that the hMSCs can successfully differentiate into multiple cell types, such as osteoblasts, chondrocytes, and adipocytes.

In addition, large-scale expansion is important for therapeutic purposes. In this study, we determined the cell proliferation rate of all isolated hMSCs. The cells were cultured until passage five. The doubling time and population doubling level (PDL) were measured for each passage. The growth profiles of all hMSCs were summarized through long-term cultivation. (S2 and S3 Tables). A direct comparison of same-donor hWJ- and hPL-MSCs demonstrated that

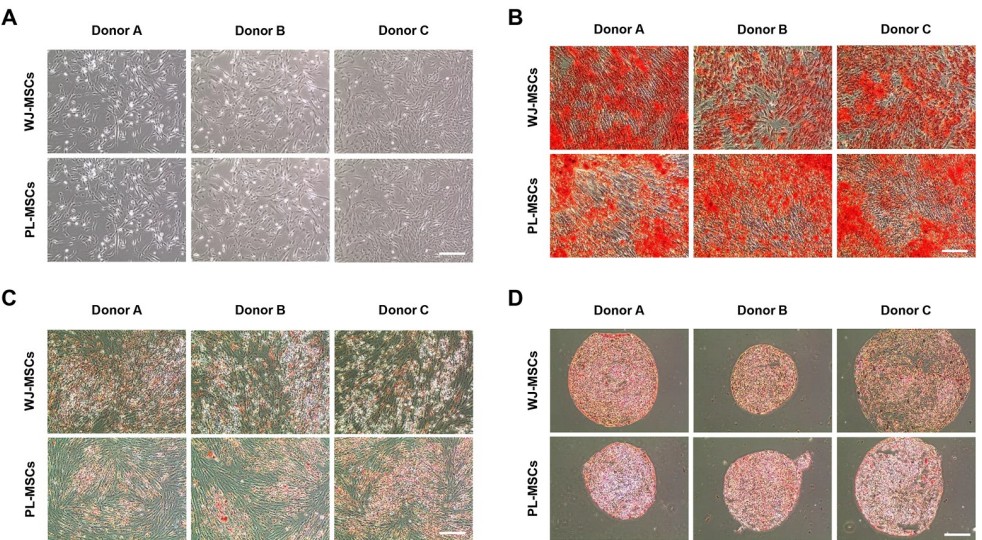

**Fig 1. Characterization of mesenchymal stem cells (MSCs) from Wharton jelly (WJ) and placenta (PL) in passage 5.** (A) Morphology of the cultured human WJ- and PL-MSCs. All hMSCs exhibited a spindle-shaped morphology. Scale bar = 100 μm; and (B-D) Multilineage differentiation was measured by staining for common lineage markers during culture in special induction media. In hMSCs from three sources, (B) osteogenic cells were evaluated through bone type alkaline phosphatase staining. (C) Adipogenic cells accumulated lipid vacuoles within the cytoplasm that stained with oil red O. (D) Chondrogenic cells accumulated sulfated proteoglycan that stained with safranin O. Scale bar = 100 μm.

the features of same-donor hMSCs were similar, without significant differences between tissue sources.

## Human WJ-MSCs prefer the anti-necroptotic pathway for anti-skeletal muscle cell death compared to PL-MSCs

To determine the difference in the anti-cell death mechanism between hWJ- and hPL-MSCs, same-donor-derived hWJ- and hPL-MSCs were co-cultured with the lovastatin-induced C2C12 cell death model. C2C12 myoblasts were pre-treated with lovastatin (10 μM) for 3 h. A 24-h co-culture was performed by inserting hWJ- and hPL-MSCs into the Transwell inserts in C2C12 cells. The co-culture was performed in serum-free media. Under co-culture for 24 h, the same-donor hMSCs from both sources inhibited cell death in C2C12 myoblasts. These results were confirmed through western blotting for apoptosis-related markers (cleaved PARP, cleaved caspase-3, cleaved caspase-8) and necroptosis-related markers (phospho-RIP3, phospho-MLKL) (Fig 2); overall, there was a consistent decrease in apoptosis and necroptosis-related marker levels. The cell death marker strength in the C2C12 cell death model co-cultured with hWJ- and hPL-MSCs was lower than that in the control group.

The apoptosis markers were significantly reduced in the group co-cultured with hPL-MSCs; while the necroptosis markers were reduced in the group co-cultured with hWJ-MSCs. This result was further confirmed using the apoptosis/necrosis detection kit. This kit analyzes each cell death state using three fluorescent dyes; calcein for cell viability (blue), apopxin for cell apoptosis (green), and 7-AAD for necrosis (red) (Fig 3). Fig 3 shows representative fluorescence images of C2C12 cells and C2C12 cell death model co-cultured with hWJ-MSCs (Fig 3A) and hPL-MSCs (Fig 3B). We observed a significant decrease in cell death in all groups co-cultured with hMSCs compared to that in the control (only C2C12 group) (Fig 3). There was a

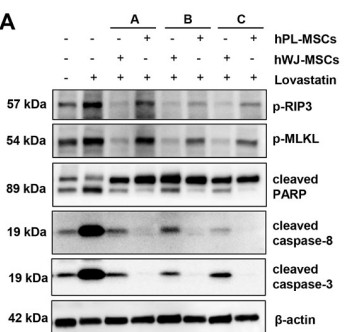
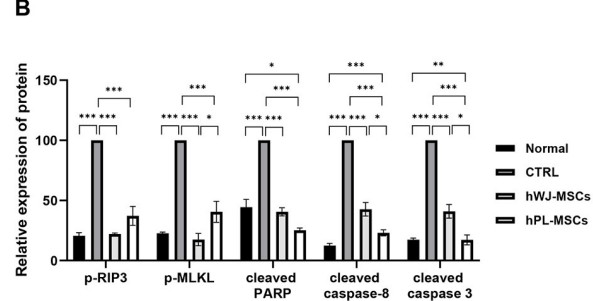

**Fig 2. Skeletal muscle cell death inhibitory mechanisms of human Wharton's jelly-derived mesenchymal stem cells (hWJ-MSCs) and human placenta derived mesenchymal stem cells (hPL-MSCs) from the same donor.** After confirming that lovastatin induced apoptosis in C2C12 cells, hWJ-MSCs (1 × 10⁵ cells/well) and hPL-MSCs (1 × 10⁵ cells/well) were co-cultured with C2C12 cells induced to undergo apoptosis with lovastatin, in a transwell chamber for 24 h. (A) Cell lysates were analyzed through western blotting using antibodies against p-RIP3, p-MLKL, cleaved PARP, cleaved caspase-8, cleaved caspase-3, and β-actin). (B) All bands were analyzed using densitometry (*** $p < 0.001$, ** $p < 0.01$, * $p < 0.05$, n = 3).

significant decrease in necrotic cells in groups co-cultured with hWJ-MSCs compared to that in cells co-cultured with hPL-MSCs. These results imply that hWJ- and hPL-MSCs can secrete soluble paracrine factors that regulate apoptosis or necroptosis in damaged C2C12 cells.

## Comparing GRO-α secretion in human WJ-MSCs and PL-MSCs that could influence anti-necroptosis in lovastatin-induced C2C12 cell death

To identify the secreted anti-necroptosis soluble factor from hWJ-MSCs, the conditioned media from each sample group, comprising hWJ- and hPL-MSCs from the same donor co-cultured with C2C12 cells, were analyzed using antibody arrays. We collected the conditioned media from the following groups to perform the array: C2C12 alone, hWJ-MSCs alone (A, B, C), hPL-MSCs alone (A, B, C), and C2C12 cells co-cultured with hWJ-MSCs (A, B, C), or hPL-MSCs (A, B, C). Comprehensive cytokine profiles were obtained from each group, and the values were normalized against a positive control; then, the protein sample content was corrected for background (Fig 4A). The secretion level of GRO-α was markedly elevated in the hWJ-MSC group co-cultured media compared to that in the hPL-MSC group co-cultured media. On the contrary, when culturing C2C12 or hPL-MSCs alone, GRO-α secretion was hardly observed (S2 Fig). The concentration of secreted GRO-α protein in each conditioned medium was measured using human GRO-α ELISA (Fig 4B). When co-cultured with C2C12 cells, the concentration increased to 22.08 ± 4.80 (PL-A) pg/mL, 20.37 ± 1.83 (PL-B) pg/mL, 11.18 ± 2.17 (PL-C) pg/mL, 1,047.35 ± 150.75 (WJ-A) pg/mL, 842.97 ± 32.99 (WJ-B) pg/mL, and 524.56 ± 13.24 (WJ-C) pg/mL. Therefore, GRO-α was selected as the secretory protein that influences anti-cell death in lovastatin-induced C2C12 cell death, based on the comparison between hWJ- and hPL-MSCs.

## Human GRO-α is a key factor in the anti-necroptotic effect of hWJ-MSCs

To determine whether GRO-α is a key factor in the anti-necroptotic effect in a skeletal muscle cell death model, *in vitro* damaged C2C12 cells were co-cultured with hWJ-MSCs. Pretreatment with GRO-α siRNAs reduced the levels of secreted GRO-α. To optimize GRO-α siRNA knockdown conditions according to specific siRNA treatment, one group of hWJ-MSCs was treated with the siGRO-α #1, and the other group was treated with siGRO-α #2 for 24 h. After

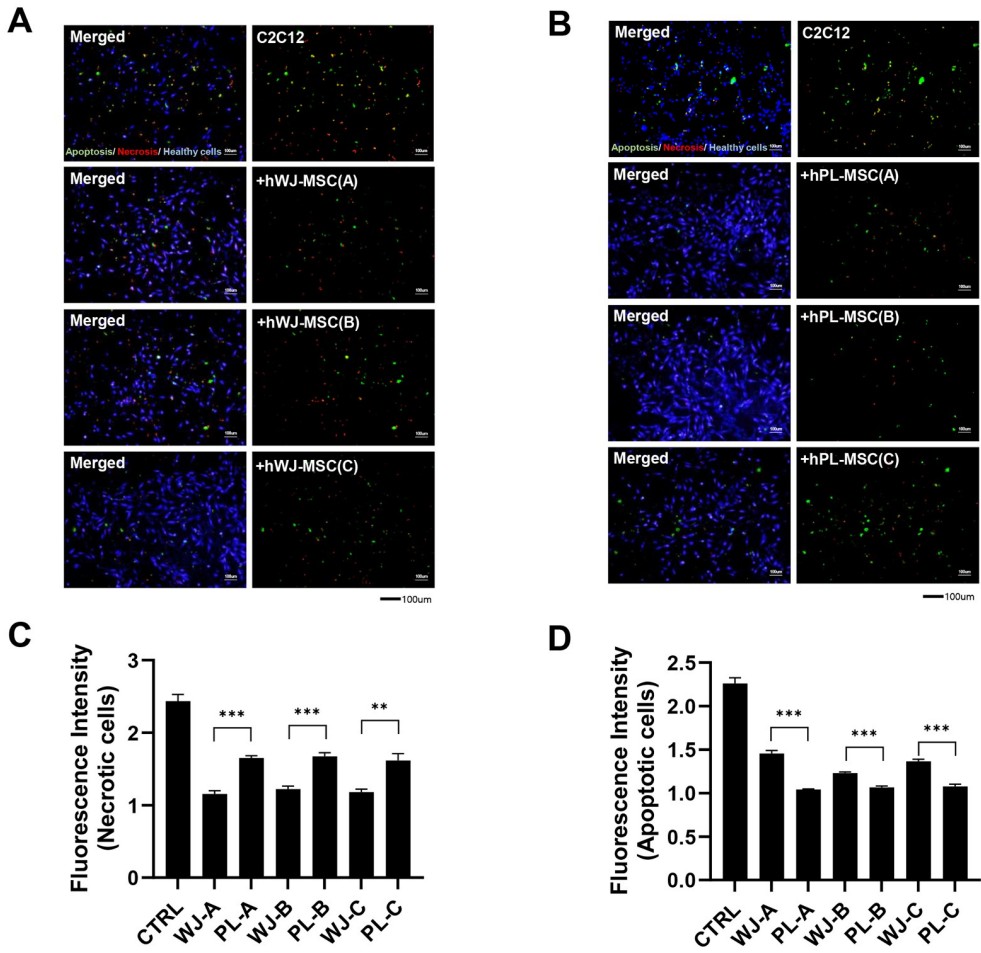

**Fig 3. Human Wharton's jelly-derived mesenchymal stem cells (hWJ-MSCs) prefer the anti-necroptotic pathway for anti-skeletal muscle cell death compared to human placenta-derived mesenchymal stem cells (hPL-MSCs) as observed through apoptosis/necrosis detection.** Lovastatin-induced C2C12 cell death was predominantly apoptotic, as determined using the apoptosis/necrosis detection kit containing apopxin green (apoptosis marker—green), 7-AAD (necrosis marker—red), and cytocalcein violet 450 (cell viability marker—blue) (scale bar = 100 μm). (A) Fluorescence image of the C2C12 cells and C2C12 cell death model co-cultured with hWJ-MSCs, (B) Fluorescence images of the C2C12 cells and C2C12 cell death model co-cultured with hPL-MSCs. (C, D) Quantitative measurements were performed using ImageJ software, and data are expressed as mean ± SEM. Data are significantly different from that in the corresponding control group. (***$p < 0.001$, **$p < 0.01$, $n = 3$, number of images in group: 2). Microscopic images (merged image) were acquired using an Olympus BX51. Microscope with a set of objective lens (Olympus Japan, UPlanFl 4x, UPlanFl 10x, UPlanFl 20x, UPlanFl 40x, UPlanFl 100x) equipped with the Olympus (DP72) Fluorescence Digital camera is used to obtain image (UPlanFl 10x) Cellsens software was used, and the resolution of the total image were measured to be 2070 × 1548 pixels, and the exposure was acquired at 200 ms.

pretreatment, the expression of the GRO-α gene and the amount of secretion into the media were analyzed (S3 Fig). To examine the knockdown of GRO-α in hWJ-MSCs, the medium was collected and GRO-α levels were analyzed using ELISA (Fig 5A). Furthermore, western blot analysis demonstrated that a knockdown of GRO-α in hWJ-MSCs reduced the anti-necrosis effects on the C2C12 cell death model. RIP-3 and MLKL phosphorylation was not attenuated in the C2C12 cell death model co-cultured with hWJ-MSCs in which GRO-α was knocked-down (Fig 5B–5D).

In addition, *in vitro* damaged C2C12 cells were treated with recombinant GRO-α proteins in a dose-dependent manner. Western blot analysis demonstrated that treatment with 50, 100,

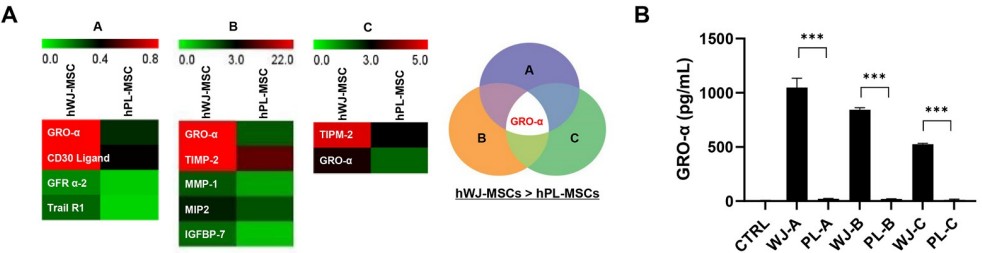

**Fig 4. Identification of growth regulated oncogene-alpha (GRO-α) secretion in human Wharton's jelly (hWJ-) and human placenta-derived mesenchymal stem cells (hPL-MSCs) that determines the anti-necroptosis in the C2C12 cell death model.** Proteins secreted from the medium were analyzed using a RayBio Biotin Label-based Human Antibody Array. (A) Increased spot intensity for GRO-α was observed in cells derived from the same donor. The selection of intersecting proteins among candidate proteins secreted by hWJ-MSCs was higher than that of hPL-MSCs. (B) The conditioned media was collected from C2C12, hWJ-MSCs alone, hPL-MSCs alone, and co-cultured cells. The concentration of secreted GRO-α in each conditioned medium was measured using the GRO-α ELISA kit (**$p < 0.01$, *$p < 0.05$, n = 4). Data are significantly different from that in the corresponding control group.

and 200 ng/mL of GRO-α was effective in reducing cell apoptosis or necroptosis. RIP-3 and MLKL phosphorylation was blocked in C2C12 cells treated with GRO-α, in a dose-dependent manner. Western blot analysis showed that the level of all cell death-related markers were reduced compared to that in the control group, in which GRO-α significantly reduced cell

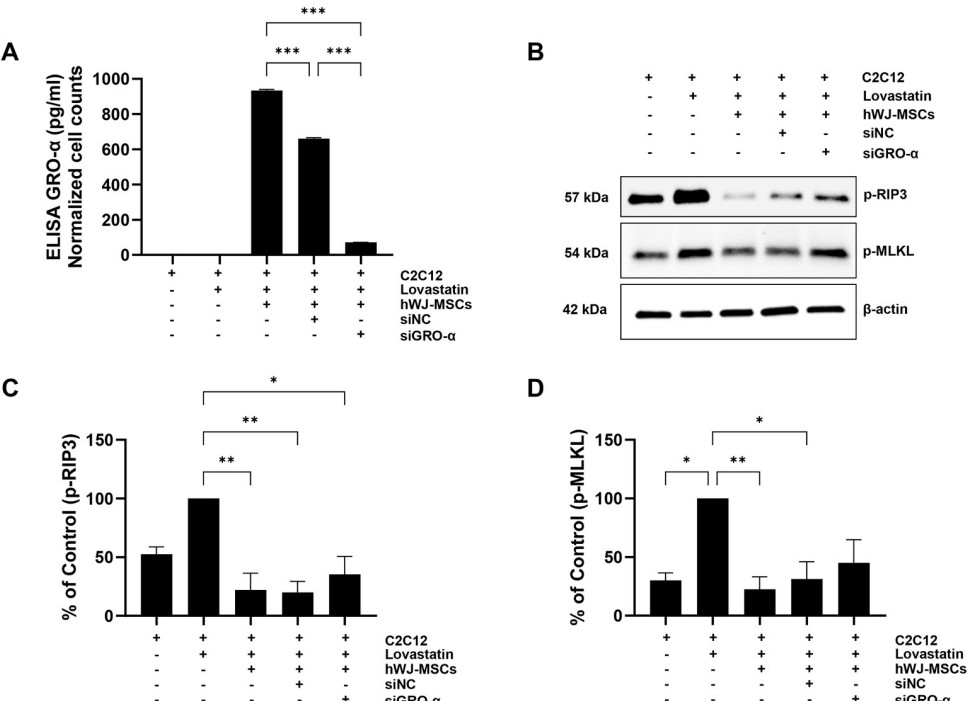

**Fig 5. Growth regulated oncogene-alpha small interfering RNA (GRO-α siRNA) treatment abolished anti-necroptotic effect of human Wharton's jelly-derived mesenchymal stem cells (hWJ-MSCs).** In vitro damaged C2C12 cells were co-cultured with hWJ-MSCs pretreated with GRO-α siRNAs, which reduced the levels of secreted GRO-α. (A) To examine knockdown of GRO-α in hWJ-MSCs, the medium was collected and GRO-α levels were analyzed using ELISA (***$p < 0.001$, n = 3). (B) Harvested cells were analyzed through western blotting with specific antibodies (p-RIP3, p-MLKL, β-actin). (C, D) All bands were analyzed through densitometry (**$p < 0.01$, *$p < 0.05$, n = 3).

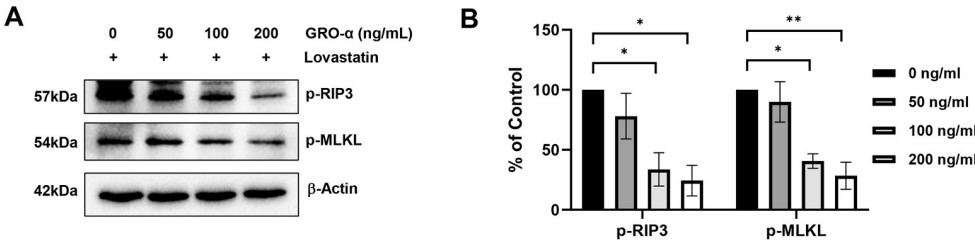

**Fig 6. Recombinant growth regulated oncogene-alpha (GRO-α) protein inhibited lovastatin-induced necroptosis of C2C12 cells.** Lovastatin inhibited cell death in C2C12 cells, both in the presence and absence of recombinant GRO-α proteins (50, 100, and 200 ng/mL) for 24 h. (A) Harvested cells were analyzed through western blotting using specific antibodies (p-RIP3, p-MLKL, β-actin). (B) All bands were analyzed using densitometry (**$p < 0.01$, *$p < 0.05$, n = 3). Data are significantly different from that in the corresponding control group.

necrosis markers compared to that in the other groups (Fig 6A and 6B). These results suggest that the secretory protein GRO-α influences necroptosis.

 To confirm the necrosis inhibitory effect of GRO-α, damaged C2C12 cells were treated with GSK'872 in a dose-dependent manner [32]. Western blotting results indicated that GSK'872 dose-dependently suppressed necroptosis; specifically, treatment with 3 and 6 nM GSK'872 was significantly effective at inhibiting necroptosis (S4 Fig). To confirm the cell necrosis suppressing efficacy of GRO-α, the cells were treated with the RIP3 inhibitor, GSK'872 and compared. Western blotting showed that the levels of all cell death-related markers were reduced in the GRO-α and GSK'872-treated groups compared to that in the control group. However, no synergistic effect was observed in the group treated with GRO-α and GSK'872 (Fig 7). These results suggest that GRO-α exerts a powerful anti-necroptotic effect by inhibiting the phosphorylation of RIP-3 and MLKL.

## hWJ-MSCs and human GRO-α inhibit necroptosis in an mdx model

The effects of hWJ-MSCs and GRO-α on Duchenne muscular dystrophy (DMD) pathology were evaluated in the gastrocnemius muscles of mdx mice. Mdx mice were intravenously (I.V.) injected with hWJ-MSCs and intraperitoneally (I.P.) injected with human GRO-α. Subsequent biochemical evaluation of muscle fiber degeneration indicated that mdx mice showed a significant increase in the creatinine kinase levels when compared to control mice. Administration of hWJ-MSCs and GRO-α significantly reduced in the mdx group compared to that in the control group (Fig 8A). We subsequently examined the localization profile of p-MLKL in

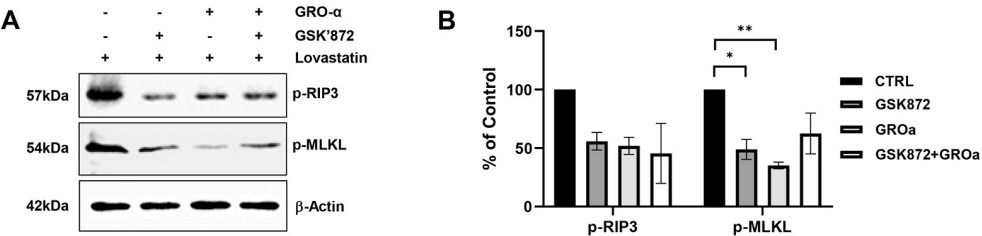

**Fig 7. Recombinant growth regulated oncogene-alpha (GRO-α) protein inhibited lovastatin-induced necroptosis of C2C12 cells more efficiently than the RIP3 inhibitor (GSK'872).** Damaged C2C12 cells were treated with GSK'872 (3 μM), GRO-α (200 ng/mL), or combined treatment (GSK'872 + GRO-α) for 24 h. (A) Harvested cells were analyzed using western blotting with antibodies against p-RIP3, p-MLKL, and β-actin. (B) All bands were analyzed using densitometry (**$p < 0.01$, *$p < 0.05$, n = 3).

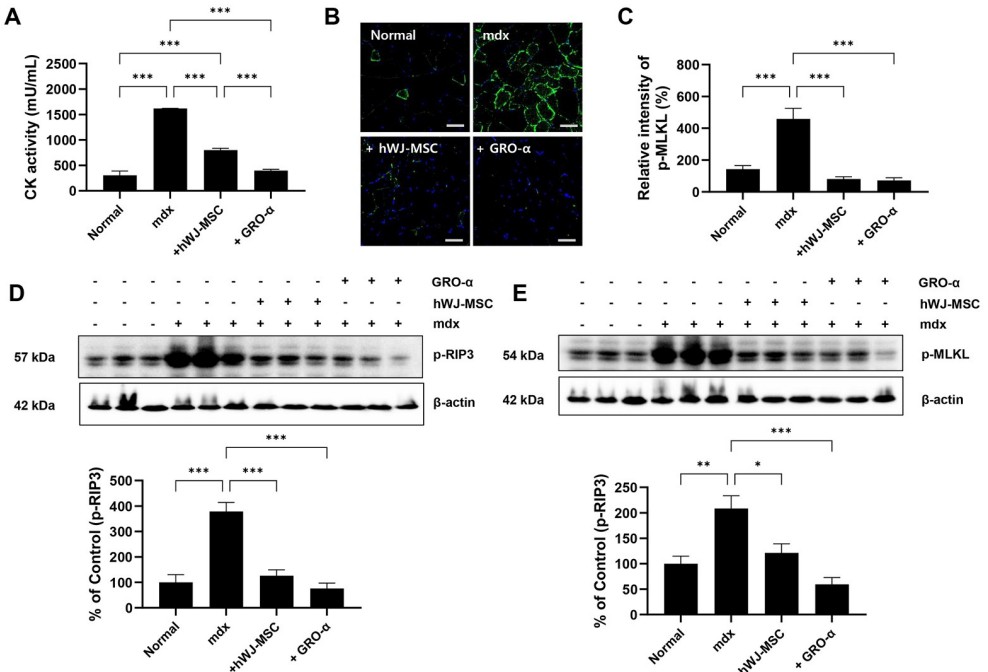

**Fig 8. Human Wharton's jelly-derived mesenchymal stem cells (hWJ-MSCs) and recombinant growth regulated oncogene-alpha (GRO-α) inhibit necroptosis in an mdx mice model.** Serum creatine kinase (CK) levels were compared among 5-month-old wild-type (NC) and mdx mice. Normal control, mdx control, hMSC-injection (mdx + hWJ-MSCs), and GRO-α (mdx + GRO-α)-treated mdx mice were used. (A) CK levels were higher in the mdx group than in the control group. The CK levels were significantly decreased in the hWJ-MSC-injected and GRO-α-treated groups. (n = 5, *** $p < 0.001$) (B) Representative immunofluorescence staining for the cell necrosis marker p-MLKL was confirmed in skeletal muscle using p-MLKL antibody under each condition. Fluorescent microscopic images (merged image) were acquired using a Zeiss confocal microscope LSM 780 (Carl Zeiss AG, Jena, Germany / Objective: Plan-Apochromat 20x/0.8 M27, Lasers: 488 nm, Cameras: Digital microscope camera AxioCam, Detectors: 2PMT, 1GaAsP, 1DIC(BF), Filter Model and Batch Number (DAPI: Ex.365, FT395, Em.445(50), GFP: Ex.470(40), FT495, Em.525(50), Rhod: Ex.546(12), FT560, Em.575-640). The image was analyzed using ZEN 2.1 software. Scale bar = 50 μm (C) Quantitative measurements were performed using ImageJ software (n = 5), and data are expressed as the mean ± SEM. Bars with different superscripts indicate significantly different values (*** $p < 0.001$, number of images in group: 3). (D) Skeletal muscle tissue lysates were analyzed through western blotting with P-RIP3 antibody. Quantitative measurements were performed using ImageJ (n = 5), and data are expressed as mean ± SEM. Bars with different superscripts indicate significantly different values (*** $p < 0.001$, ** $p < 0.01$). (E) Skeletal muscle tissue lysates were analyzed through western blotting with p-MLKL antibody. Quantitative measurements were performed using ImageJ software (n = 5), and data are expressed as mean ± SEM. Bars with different superscripts indicate significantly different values (*** $p < 0.001$, ** $p < 0.01$, * $p < 0.05$).

dystrophic muscles using immunofluorescence. Areas with strong sarcoplasmic immunoreactivity to the p-MLKL antibody was observed in the dystrophic muscles of mdx mice; these were not observed in the control groups (Fig 8B and 8C). Western blot analysis of the gastrocnemius muscle tissue revealed significantly increased expression of the necrosis markers, p-MLKL and p-RIP3 in the mdx group compared to the normal control group; these markers were significantly downregulated in the GRO-α and hWJ-MSCs treated groups (Fig 8D and 8E). In agreement with the *in vitro* results, this confirmed that necroptosis was reduced by injection of GRO-α and hWJ-MSCs in a DMD mouse model.

## Discussion

hMSCs are paracrine, multipotent stem cell populations that have clinical applications [33,34]. As hMSCs can be obtained from various tissues, tissue selection is an important factor when

developing stem cell therapeutics [10,35,36]. Portions of hMSCs from birth-associated tissues, preferably placenta and WJ, bring advantages such as non-invasiveness and ethical problem-free availability [26]. In addition, MSCs from these birth-associated tissues exhibit an enhanced proliferative capacity compared to some hMSC populations obtained from adult tissues [24,37]. Thus, human birth-associated tissue-derived hMSCs are promising candidates for cell therapy. Nevertheless, there are no comparative studies on the therapeutic efficacy or characteristics of birth-associated tissue-derived MSCs (hPL-, hWJ-, and hUCB-MSCs) from the same donor. In this study, we isolated hWJ- and hPL-MSCs from the same donor and compared their morphology, doubling time, trilineage differentiation potential, and stemness-related surface markers. Our previous studies confirmed that hWJ- and hPL-MSCs have the potential to inhibit skeletal muscle cell death compared to hAT- or hBM-MSCs [27]. Therefore, in this study, the anti-skeletal muscle cell death mechanisms of the same donor-derived hWJ- and hPL-MSCs were compared.

Same donor-derived hWJ- and hPL-MSCs exhibited similar hMSC morphologies, doubling time, trilineage differentiation potential, and stemness-related surface markers. However, despite their similar characteristics, hWJ-MSCs had a higher ability to inhibit skeletal muscle cell necroptosis than hPL-MSCs. By contrast, hPL-MSCs had a higher ability to inhibit skeletal muscle cell apoptosis than hWJ-MSCs. In particular, the preference of anti-necroptosis varied; the secretion level of GRO-α was markedly elevated in hWJ-MSCs co-cultured media, compared to that in hPL-MSCs co-cultured media. hMSCs from different sources exhibit, epigenetic and transcriptomic differences based on the source tissues [27,38,39]. The angiogenic potential of WJ- and BM-derived MSCs is higher than that of AT-MSCs [40]. The characteristics vary with the source; the secretion of VEGF and TGF-β1 is higher and comparable in the amniotic fluid and membrane (AM)-MSCs and AT-MSCs, while their levels are lower in UCB-MSCs, with an associated opposing expression trend for HGF [41,42]. The secretion of VEGF-A, HGF, bFGF, and ANG-1 was higher in amniotic-derived MSCs than in AT-MSCs [43]. Furthermore, the expression of macrophage colony-stimulating factor (M-CSF), interleukin-1 receptor antagonist (IL-1ra), and SDF-1a secretion was significantly higher in TNF-α- and IFN-γ-stimulated perinatal MSCs; but lower than in BM-MSCs. The expression of monocyte chemotactic protein-1 (MCP-1) was significantly higher in perinatal MSCs than in BM-MSCs with without changes after stimulation [42]. Our results confirmed that although the characteristics of hMSCs are similar, the mechanism by which they inhibit cell death differs depending on the tissue of origin.

GRO-α was differentially expressed in hWJ- and hPL-MSCs. GRO-α is a member of the CXC family. It is a chemoattractant for several immune cells, particularly neutrophils or other non-hematopoietic cells, at the site of injury or infection. It plays an important role in regulating immune and inflammatory responses [44–46]. GRO-α also plays an important role in spinal cord development by inhibiting the migration of oligodendrocyte precursors. It is involved in angiogenesis, inflammation, wound healing, and tumor formation [47–49]. The anti-necroptotic role of GRO-α has not been reported, even in well-known immune systems. In this study, to determine whether GRO-α is a critical factor mediating the anti-necroptotic effects of hWJ-MSCs, skeletal muscle cells were treated with recombinant GRO-α protein in the presence of lovastatin. Treatment with recombinant GRO-α dramatically protected the cells from lovastatin-induced C2C12 cell death through necrosis inhibition. In addition, the anti-necroptotic effect was higher than that with GSK'872, an effective necroptosis inhibitor. Therefore, GRO-α is a powerful anti-necroptotic that inhibits the phosphorylation of RIP-3 and MLKL. Nonetheless, no synergistic effect was observed in the GRO-α and GSK'872 groups. Further studies are required to determine whether the mechanisms by which GRO-α and GSK'872 inhibit necroptosis differ.

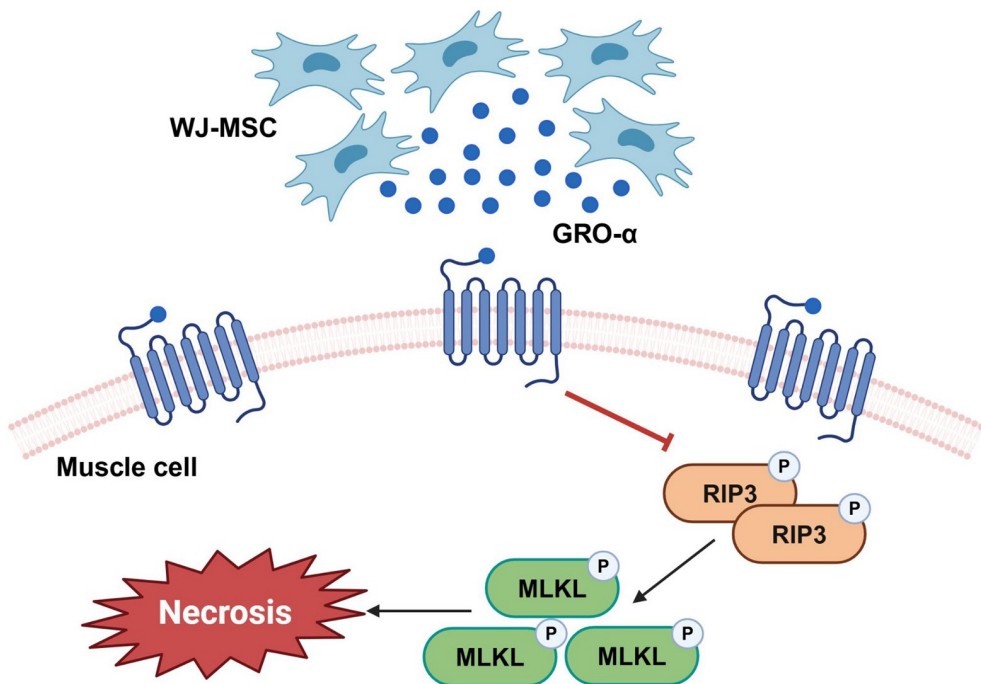

**Fig 9. Anti-necroptotic effects of human Wharton's jelly-derived mesenchymal stem cells in skeletal muscle cell death model via secretion of growth regulated oncogene-alpha (GRO-α).** Exposure of WJ-MSCs to the damaged skeletal myocyte environment promotes the secretion of GRO-α. In addition, GRO-α plays a role in preventing skeletal muscle cell damage by influencing the anti-necrotic pathway while inhibiting phosphorylation of RIP-3 and MLKL. Graphical illustrations presented in the present study were generated using BioRender (https://www.biorender.com/).

We could not confirm the influence of sex of the hWJ-MSCs donor [50]; however, our findings demonstrated that hWJ-MSCs and the derived GRO-α are a class of novel necroptosis inhibitors that can selectively suppress RIP3 and MLKL activities and potently block necroptotic cell death.

Apoptosis and necroptosis are major mechanisms of cell death [51–53]. Apoptosis is widely recognized as the major form of programmed cell death, which is triggered by various death signals such as free radical production, triggering the activation of caspases and sequential biochemical reactions [54]. Necroptosis is a newly discovered pathway of regulated and programmed necrosis involving the activity of receptor-interacting protein kinase 3 (RIPK3) and mixed lineage kinase domain-like protein (MLKL), which has important kinase-dependent functions that inhibit or induce necroptosis [53]. Necroptosis is known to be involved in the pathogenesis of disorders that can affect tissues including the brain, heart, liver, kidneys, and pancreas [55,56]. Several recent studies have reported that necroptosis is involved in skeletal muscle apoptosis diseases such as DMD, particularly in the degeneration of dystrophin-deficient muscles [57,58]. Necroptosis contributes to myofiber death in dystrophin-deficient skeletal muscles by activating RIPK3-dependent necroptotic signaling [59]. DMD is a fatal X-linked recessive disorder characterized by the progressive loss of muscle mass and function. It occurs in approximately 1/3,500 live male births, and is caused by deleterious mutations in the dystrophin gene [60–62]. Currently, treatment strategies for DMD are limited, and there is no specific treatment for muscle cell death [63]. Our results confirmed that hWJ-MSCs and derived GRO-α have the potential to become an effective DMD therapeutic agent through inhibition of necroptosis.

## Conclusion

Human hWJ-MSCs, which could be effective therapeutic agents for muscle diseases, exert anti-apoptotic, anti-inflammatory, and immunomodulatory effects on surrounding cells through the secretion of paracrine factors. To the best of our knowledge, this is the first study to compare the characteristics and anti-skeletal muscle cell death mechanisms of hWJ- and hPL-MSCs derived from the same donor. The exposure of WJ-MSCs to the damaged skeletal myocyte environment promotes the secretion of GRO-α. GRO-α plays a role in preventing skeletal muscle cell damage by influencing the anti-necrotic pathway while inhibiting phosphorylation of RIP-3 and MLKL (Fig 9). Our findings demonstrate that by inhibiting necroptosis in skeletal muscle cells, hWJ-MSCs and derived GRO-α are novel treatment options for skeletal muscle diseases such as DMD.

## Supporting information

**S1 Fig. The original images underlying all western blotting results.**
(PDF)

**S2 Fig. Representative scan images of antibody array.** The yellow box represents growth regulated oncogene-alpha (GRO-α) expression in the culture medium under each condition.
(TIF)

**S3 Fig. Optimization of growth regulated oncogene-alpha small interfering RNA (GRO-α siRNA) knockdown condition in Wharton's jelly-derived human mesenchymal stem cells (hWJ-MSCs).** To knockdown GRO-α in hWJ-MSCs, hWJ-MSCs were pretreated with an siRNA control (siNC) or siRNA #1 and siRNA #2 against GRO-α for 24 and 48h (A, B) The GRO-α mRNA expression levels in hWJ-MSCs were measured using qRT-PCR (\*\*\*$p < 0.001$; n = 3). (C) The GRO-α levels in each media were measured using ELISA (\*\*\*$p < 0.001$; n = 3).
(TIF)

**S4 Fig. Dose-dependent anti-necrotic effect of RIP kinase inhibitor (GSK'872).** To confirm the GSK'872 concentration for anti-necrosis, the damaged C2C12 cells were treated with various GSK'872 concentrations. (A) The harvested cells were analyzed using western blotting with antibodies against p-RIP3, p-MLKL, and β-actin. (B) All bands were analyzed by densitometry (\*\*\*$p < 0.001$, \*$p < 0.05$, n = 3). These results were significantly different from those of the corresponding control group.
(TIF)

**S1 Table. Stemness validation of human mesenchymal stem cells (hMSCs).** The hMSC surface markers were identified using flow cytometry. The results for positive (CD44, CD73, CD90, and CD105) and negative (CD11b, CD14, CD19, CD34, CD45, and HLA-DR) markers are shown.
(DOCX)

**S2 Table. Doubling time for human mesenchymal stem cells (hMSCs).**
(DOCX)

**S3 Table. Cumulative population doubling level for human mesenchymal stem cells (hMSCs).**
(DOCX)

## Author Contributions

**Conceptualization:** Hong Bae Jeon, Jong Wook Chang.

**Data curation:** Sang Eon Park, Sun Jeong Kim, Min-Jeong Kim, Soo-young Oh, Gyu Ha Ryu.

**Formal analysis:** Soo Jin Kwon, Jang Bin Jeong.

**Funding acquisition:** Jong Wook Chang.

**Investigation:** Soo Jin Kwon.

**Methodology:** Sang Eon Park, Soo Jin Kwon, Jang Bin Jeong, Min-Jeong Kim, Suk-joo Choi.

**Project administration:** Sang Eon Park.

**Resources:** Suk-joo Choi, Soo-young Oh.

**Supervision:** Hong Bae Jeon, Jong Wook Chang.

**Validation:** Sun Jeong Kim.

**Visualization:** Sun Jeong Kim.

**Writing – original draft:** Sang Eon Park, Soo Jin Kwon, Hong Bae Jeon, Jong Wook Chang.

**Writing – review & editing:** Sun Jeong Kim, Hong Bae Jeon, Jong Wook Chang.

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
