## [Decision Letter · Decision Letter 0]

6 Sep 2024

PONE-D-24-11604Anti-necroptotic effects of human Wharton’s jelly-derived mesenchymal stem cells in skeletal muscle cell death model via secretion of GRO-αPLOS ONE

Dear Dr. Jeon,

Thank you for submitting your manuscript to PLOS ONE. After careful consideration, we feel that it has merit but does not fully meet PLOS ONE’s publication criteria as it currently stands. Therefore, we invite you to submit a revised version of the manuscript that addresses the points raised during the review process.

We look forward to receiving your revised manuscript.

Kind regards,

Atsushi Asakura, Ph.D

Academic Editor

PLOS ONE

Journal requirements: 1. When submitting your revision, we need you to address these additional requirements. Please ensure that your manuscript meets PLOS ONE's style requirements, including those for file naming. The PLOS ONE style templates can be found at https://journals.plos.org/plosone/s/file?id=wjVg/PLOSOne_formatting_sample_main_body.pdf and https://journals.plos.org/plosone/s/file?id=ba62/PLOSOne_formatting_sample_title_authors_affiliations.pdf. 2. PLOS ONE now requires that authors provide the original uncropped and unadjusted images underlying all blot or gel results reported in a submission’s figures or Supporting Information files. This policy and the journal’s other requirements for blot/gel reporting and figure preparation are described in detail at https://journals.plos.org/plosone/s/figures#loc-blot-and-gel-reporting-requirements and https://journals.plos.org/plosone/s/figures#loc-preparing-figures-from-image-files. When you submit your revised manuscript, please ensure that your figures adhere fully to these guidelines and provide the original underlying images for all blot or gel data reported in your submission. See the following link for instructions on providing the original image data: https://journals.plos.org/plosone/s/figures#loc-original-images-for-blots-and-gels.   In your cover letter, please note whether your blot/gel image data are in Supporting Information or posted at a public data repository, provide the repository URL if relevant, and provide specific details as to which raw blot/gel images, if any, are not available. Email us at plosone@plos.org if you have any questions. 2. PLOS requires an ORCID iD for the corresponding author in Editorial Manager on papers submitted after December 6th, 2016. Please ensure that you have an ORCID iD and that it is validated in Editorial Manager. To do this, go to ‘Update my Information’ (in the upper left-hand corner of the main menu), and click on the Fetch/Validate link next to the ORCID field. This will take you to the ORCID site and allow you to create a new iD or authenticate a pre-existing iD in Editorial Manager. 3. We note that the grant information you provided in the ‘Funding Information’ and ‘Financial Disclosure’ sections do not match.  When you resubmit, please ensure that you provide the correct grant numbers for the awards you received for your study in the ‘Funding Information’ section. 4. We note that you have included the phrase “data not shown” in your manuscript. Unfortunately, this does not meet our data sharing requirements. PLOS does not permit references to inaccessible data. We require that authors provide all relevant data within the paper, Supporting Information files, or in an acceptable, public repository. Please add a citation to support this phrase or upload the data that corresponds with these findings to a stable repository (such as Figshare or Dryad) and provide and URLs, DOIs, or accession numbers that may be used to access these data. Or, if the data are not a core part of the research being presented in your study, we ask that you remove the phrase that refers to these data. 5. Your ethics statement should only appear in the Methods section of your manuscript. If your ethics statement is written in any section besides the Methods, please move it to the Methods section and delete it from any other section. Please ensure that your ethics statement is included in your manuscript, as the ethics statement entered into the online submission form will not be published alongside your manuscript.  6. Please remove your figures from within your manuscript file, leaving only the individual TIFF/EPS image files, uploaded separately. These will be automatically included in the reviewers’ PDF. 7. Please include captions for your all Supporting Information files at the end of your manuscript, and update any in-text citations to match accordingly. Please see our Supporting Information guidelines for more information: http://journals.plos.org/plosone/s/supporting-information. 

Reviewers' comments:

Reviewer's Responses to Questions

**Comments to the Author**

1. Is the manuscript technically sound, and do the data support the conclusions?

Reviewer #1: Yes

2. Has the statistical analysis been performed appropriately and rigorously? 

Reviewer #1: Yes

3. Have the authors made all data underlying the findings in their manuscript fully available?

Reviewer #1: Yes

4. Is the manuscript presented in an intelligible fashion and written in standard English?

Reviewer #1: Yes

5. Review Comments to the Author

Reviewer #1: PONE-D-24-11604

Comments to the Editor

There is a sizeable literature covering the impact of site of origin of human mesenchymal stem cells (hMSCs) on their properties and potential as therapeutic agents for multiple conditions. The Authors have extended their work in this area by comparing the effects of h-MSCs isolated from placenta and Wharton’s jelly from the same individuals on in vivo and in vitro models of skeletal muscle cell death. They find that secretions from hWJ-MSCs were highly effective in inhibiting necroptosis and provide evidence suggesting that the primary mediator of this effect is GRO� release by hWJ-MSCs.

The manuscript has several strengths. One is the use of multiple markers of pathways of cell death. There is also a comprehensive analysis of the basic properties of the hMSCs from the two sources, mirroring the analysis presented in their earlier work (ref #27). Most important is that they compare PL and WJ-derived cells from the same subjects, even if only 3 different individuals are studied.

There are several potential major problems. One is the low number of replicate determinations for each of the sets of results (n=3). This would usually represent the bare minimum acceptable number for a communication, not a full paper. Yet, in this instance it need not be considered a disqualifying factor, as the treatment effects are such a magnitude that additional replicates would not make the case they are presenting much stronger. A second one is that in the current manuscript they report that when hMSCs are cultured alone, GRO� is not detectable in the media, yet in their previous work (ref #27, Fig. 7e) it appears that solo-cultured hSMCs release appreciable amounts of CXCL1/GRO�. This apparent discrepancy needs to be discussed; might it be due to variability between donors? Or the sensitivity of the detection methodology?

Comments to the Authors

There is a sizeable literature covering the impact of site of origin of human mesenchymal stem cells (hMSCs) on their properties and potential as therapeutic agents for multiple conditions. The Authors have extended their work in this area by comparing the effects of h-MSCs isolated from placenta and Wharton’s jelly from the same individuals on in vivo and in vitro models of skeletal muscle cell death. They find that secretions from hWJ-MSCs were highly effective in inhibiting necroptosis and provide evidence suggesting that the primary mediator of this effect is GRO� release by hWJ-MSCs.

General comments.

1. In the current manuscript the Authors report that when hMSCs are cultured alone, GRO� is not detectable in the media, yet in their previous work (ref #27, Fig. 7e) it appears that solo-cultured hMSCs release appreciable amounts of CXCL1/GRO�. This apparent discrepancy needs to be discussed; might it be due to variability between donors? Or the sensitivity of the detection methodology? It is an important point, as it impacts their hypothesis that some aspect of the microenvironment of challenged C2C12 cells is required to induce GRO� secretion by hWJ-MSCs.

Specific comments

1. Abstract, L.40-41. The first mention of GRO� is somewhat confusing. It might be better to move the sentence starting “GRO� was differentially….” (L.43) to earlier in the abstract, setting up why a focus was applied to GRO�.

2. Section 2.2, Co-culture. The sequence of these events is somewhat unclear. Were C2C12 cells reseeded after exposure to lovastatin (L. 94)? Was lovastatin still present during co-culture?

3. There is some confusion about the method(s?) of euthanasia. Was it isoprene (l.114), or CO2 (L.116-119)?

4. L.129. Please provide a reference for the cell counting methodology.

5. Section 2.814. Please provide catalog numbers for the antibodies used.

6. L. 245. “…features similar…” is lacking in detail. Please expand on what the Authors mean.

7. L. 311. Provide the source and performance characteristics of the GRO� ELISA, especially the lower limit of detection. See General comment.

8. Fig. 5.

6. PLOS authors have the option to publish the peer review history of their article (what does this mean?). If published, this will include your full peer review and any attached files.

Reviewer #1: **Yes: **Theodore P Ciaraldi

---

## [Author Response · Author response to Decision Letter 0]

25 Sep 2024

Journal requirements_1:

1. In your Data Availability statement you say that the antibody array data generated in the present study can be found in the GEO under accession number GSE277126 (https://www.ncbi.nlm.nih.gov/geo/query/acc.cgi?acc=GSE277126). Accession "GSE277126" is currently private and is scheduled to be released on Dec 31, 2024. Can you update this so that the data can be viewed.

Reply: The antibody array data generated in the present study can be found in the GEO under the accession number GSE277126 (https://www.ncbi.nlm.nih.gov/geo/query/acc.cgi?acc=GSE277126). As your request, we have updated the data release date to September 25, 2024.

Journal requirements_2:

 Reply: We have modified our manuscript to meet PLOS ONE’s style requirements, including those for file naming. 

 Reply: According to the journal’s request, we have provided the original uncropped and unadjusted images underlying all blot or gel results reported in the Supporting Information file S4 Fig. In addition, we have also highlighted this in the cover letter. 

 Reply: We have authenticated and added the ORCID iD of the corresponding author.

 Reply: To match ‘Funding information’ and ‘Financial disclosure’, we have included the changed financial disclosures in the cover letter and provided the correct grant numbers for the awards we received for our study in the ‘Funding information’ section. 

 Reply: The results have been submitted as Supporting Information (S3 Fig).

 Reply: We have moved the ethics statement to the Materials and methods section of the manuscript.

6. Please remove your figures from within your manuscript file, leaving only the individual TIFF/EPS image files, uploaded separately. These will be automatically included in the reviewers’ PDF.

 Reply: We have deleted the figures in the manuscript and uploaded the individual TIFF image files separately.

7. Please include captions for your all Supporting Information files at the end of your manuscript, and update any in-text citations to match accordingly. Please see our Supporting Information guidelines for more information: http://journals.plos.org/plosone/s/supporting-information. 

Reply: Referring to the Support Information Guidelines, we have included captions of all support information files at the end of the manuscript and updated the in-text citations to match them accordingly.

Reviewer #1: PONE-D-24-11604

Comments to the Authors

There is a sizeable literature covering the impact of site of origin of human mesenchymal stem cells (hMSCs) on their properties and potential as therapeutic agents for multiple conditions. The Authors have extended their work in this area by comparing the effects of h-MSCs isolated from placenta and Wharton’s jelly from the same individuals on in vivo and in vitro models of skeletal muscle cell death. They find that secretions from hWJ-MSCs were highly effective in inhibiting necroptosis and provide evidence suggesting that the primary mediator of this effect is GROalpha release by hWJ-MSCs.

Reply: We appreciate the reviewer’s constructive review of the manuscript. We believe that the reviewer has raised an important issue and agree with this comment.

General comments.

1. In the current manuscript the Authors report that when hMSCs are cultured alone, GRO alpha is not detectable in the media, yet in their previous work (ref #27, Fig. 7e) it appears that solo-cultured hMSCs release appreciable amounts of CXCL1/GRO alpha. This apparent discrepancy needs to be discussed; might it be due to variability between donors? Or the sensitivity of the detection methodology? It is an important point, as it impacts their hypothesis that some aspect of the microenvironment of challenged C2C12 cells is required to induce GRO alpha secretion by hWJ-MSCs.

Reply: Thank you for your comment. We apologize for the confusion caused by what we wrote. We have added Fig. S1 for a clear understanding of the content. As shown in the figure, fluorescence in the antibody array was hardly detected in C2C12 alone or hPL-MSCs alone. Unfortunately, the part that was not detected in the hWJ-MSC only is a typo. As shown in Fig. S2B, GROα secretion was confirmed in hWJ-MSC alone. Therefore, we have modified this sentence accordingly.

Line 312-313. On the contrary, when culturing C2C12 or hPL-MSCs alone, GRO-α secretion was hardly observed (S1 Fig).

Specific comments

1. Abstract, L.40-41. The first mention of GRO alpha is somewhat confusing. It might be better to move the sentence starting “GRO alpha was differentially….” (L.43) to earlier in the abstract, setting up why a focus was applied to GRO alpha.

Reply: We agree with the reviewer’s comment and have revised the sentence accordingly.

Line 25-37. Human mesenchymal stem cells (hMSCs) have therapeutic applications and potential for use in regenerative medicine. However, the use of hMSCs in research and clinical medicine is limited by a lack of information pertaining to their donor-specific functional attributes. In this study, we compared the characteristics of same-donor derived placenta (PL) and Wharton’s jelly (WJ)-derived hMSCs, we also compared their mechanism of action in a skeletal muscle disease in vitro model. The same-donor-derived hWJ- and hPL-MSCs exhibited typical hMSC characteristics. However, GRO-α was differentially expressed in hWJ- and hPL-MSCs. hWJ-MSCs, which secreted a high amount of GRO-α, displayed a higher ability to inhibit necroptosis in skeletal muscle cells than hPL-MSCs. This demonstrates the anti-necroptotic therapeutic effect of GRO-α in the skeletal muscle cell death model. Furthermore, GRO-α also exhibited the anti-necroptotic effect in a Duchenne muscular dystrophy (DMD) mouse model. Considering their potential to inhibit necroptosis in skeletal muscle cells, hWJ-MSCs and the derived GRO-α are novel treatment options for skeletal muscle diseases such as DMD.

2. Section 2.2, Co-culture. The sequence of these events is somewhat unclear. Were C2C12 cells reseeded after exposure to lovastatin (L. 94)? Was lovastatin still present during co-culture?

Reply: We revised the sentence to clarify the content. The modified part now reads as follows:

Line 85-90. When C2C12 reached 80–90% confluency after seeding on a six-well plate (BD Falcon, Franklin Lakes, NJ, USA), they were treated with 10 μM lovastatin for 3 h to induce cell death and washed with PBS to remove lovastatin. hWJ- and hPL-MSCs were trypsinized, suspended in serum-free MEM alpha, seeded (1×105/1 mL) in 1-µm-pore-size transwell inserts (BD Biosciences, Franklin Lakes, NJ, USA), and co-cultured with C2C12 for 24 h.

3. There is some confusion about the method(s?) of euthanasia. Was it isoprene (l.114), or CO2 (L.116-119)?

Reply: We apologize for the confusion regarding the euthanasia method used. Mice were euthanized using carbon dioxide, not isoprene. So, we deleted the sentence “The mice were euthanized using isoprene, 7 days after administration”.

4. L.129. Please provide a reference for the cell counting methodology.

Reply: As you advised, we have provided a reference for the cell counting methodology.

Line 119-120. The viability of C2C12 cells was measured using the Cell Counting kit-8 (CCK-8, Dojindo, Tokyo, Japan), as previously described [30].

5. Section 2.814. Please provide catalog numbers for the antibodies used.

Reply: As you requested, we provided catalog numbers for antibodies used. The modified lines are as follows:

Line 185-191. The following primary antibodies were used: poly ADP ribose polymerase (9542L, PARP; polyclonal, Cell Signaling Technology, Danvers, MA, USA), phosphoreceptor interacting protein kinase 3 (57220S, P-RIP3; polyclonal, Cell Signaling Technology), phospho-mixed lineage kinase domain-like protein (37333S, P-MLKL; monoclonal (D6E3G), Cell Signaling Technology), cleaved Caspase-3 (9661S, polyclonal, Cell Signaling Technology), and beta-actin (sc-47778, monoclonal (C4), Santa Cruz Biotechnology, Santa Cruz, CA, USA).

6. L. 245. “…features similar…” is lacking in detail. Please expand on what the Authors mean.

Reply: As you advised, we expanded the sentence to clarify the meaning.

Line 245-247. A direct comparison of same-donor hWJ- and hPL-MSCs demonstrated that the features of same-donor hMSCs were similar, without significant differences between tissue sources.

7. L. 311. Provide the source and performance characteristics of the GRO alpha ELISA, especially the lower limit of detection. See General comment.

Reply: We used a human CXCL1/GRO alpha ELISA kit (DGR00B). The sensitivity of this is 10 pg/ml and assay range is 31.3 - 1,000 pg/ml. Sample types allowed this kit are cell culture supernatants, serum, platelet-poor EDTA plasma, platelet-poor heparin plasma, and platelet-poor citrate plasma. Specificity is natural and recombinant human GRO-alpha. A cross-reactivity of < 0.5% was observed for the available related molecules. Cross-specific reactivity of < 50% was observed for the species tested. No significant interference was observed with the available related molecules.

8. Fig. 5.

Reply: As shown in Fig 5A, normal and lovastatin-treated C2C12 cells did not release GRO-alpha. This is because a human GRO alpha ELISA kit was used to confirm GRO-alpha secretion. However, the hWJ-MSCs alone released a significant amount of GRO-alpha (Fig S2B).

---

## [Decision Letter · Decision Letter 1]

30 Oct 2024

Anti-necroptotic effects of human Wharton’s jelly-derived mesenchymal stem cells in skeletal muscle cell death model via secretion of GRO-α

PONE-D-24-11604R1

Dear Dr. Jeon,

We’re pleased to inform you that your manuscript has been judged scientifically suitable for publication and will be formally accepted for publication once it meets all outstanding technical requirements.

Kind regards,

Atsushi Asakura, Ph.D

Academic Editor

PLOS ONE

Additional Editor Comments (optional):

Reviewers' comments:

Reviewer's Responses to Questions

**Comments to the Author**

1. If the authors have adequately addressed your comments raised in a previous round of review and you feel that this manuscript is now acceptable for publication, you may indicate that here to bypass the “Comments to the Author” section, enter your conflict of interest statement in the “Confidential to Editor” section, and submit your "Accept" recommendation.

Reviewer #1: All comments have been addressed

2. Is the manuscript technically sound, and do the data support the conclusions?

Reviewer #1: Yes

3. Has the statistical analysis been performed appropriately and rigorously? 

Reviewer #1: Yes

4. Have the authors made all data underlying the findings in their manuscript fully available?

Reviewer #1: Yes

5. Is the manuscript presented in an intelligible fashion and written in standard English?

Reviewer #1: Yes

6. Review Comments to the Author

Reviewer #1: (No Response)

7. PLOS authors have the option to publish the peer review history of their article (what does this mean?). If published, this will include your full peer review and any attached files.

Reviewer #1: **Yes: **Theodore P Ciaraldi

---

## [Editor Report · Acceptance letter]

19 Nov 2024

PONE-D-24-11604R1 

PLOS ONE

Dear Dr. Jeon, 

I'm pleased to inform you that your manuscript has been deemed suitable for publication in PLOS ONE. Congratulations! Your manuscript is now being handed over to our production team.

Kind regards, 

on behalf of

Dr. Atsushi Asakura 

Academic Editor

PLOS ONE